# Functional Food Product Based on Nanoselenium-Enriched *Lactobacillus casei* against Cadmium Kidney Toxicity

**Simona Ioana Vicas** [1,†], **Vasile Laslo** [1,†], **Adrian Vasile Timar** [1,†], **Cornel Balta** [2,†], **Hildegard Herman** [2,†], **Alina Ciceu** [2,†], **Sami Gharbia** [2,†], **Marcel Rosu** [2,†], **Bianca Mladin** [2,†], **Luminita Fritea** [3,†], **Simona Cavalu** [3,†], **Coralia Cotoraci** [4,†], **József Prokisch** [5,†], **Maria Puschita** [4,†], **Calin Pop** [4,†], **Eftimie Miutescu** [4,†] and **Anca Hermenean** [2,4,*,†]

1   Faculty of Environmental Sciences, University of Oradea, 410048 Oradea, Romania; svicas@uoradea.ro (S.I.V.); vlaslo@uoradea.ro (V.L.); atimar@uoradea.ro (A.V.T.)
2   "AurelArdelean" Institute of Life Sciences, Vasile Goldis Western University of Arad, 310414 Arad, Romania; baltacornel@gmail.com (C.B.); hildegard.i.herman@gmail.com (H.H.); alinaciceu80@gmail.com (A.C.); samithgh2@hotmail.com (S.G.); ramrosu@gmail.com (M.R.); biancaonitamaria@gmail.com (B.M.)
3   Faculty of Medicine and Pharmacy, University of Oradea, 410073 Oradea, Romania; fritea_luminita@yahoo.com (L.F.); simona.cavalu@gmail.com (S.C.)
4   Faculty of Medicine, Vasile Goldis Western University of Arad, 310414 Arad, Romania; cotoraci.coralia@uvvg.ro (C.C.); mpuschita@yahoo.com (M.P.); medicbm@yahoo.com (C.P.); miucescu.eftimie@uvvg.ro (E.M.)
5   Institute of Animal Science, Biotechnology and Nature Conservation, Faculty of Agricultural and Food Sciences and Environmental Management, University of Debrecen, 4032 Debrecen, Hungary; jprokisch@agr.unideb.hu
*   Correspondence: anca.hermenean@gmail.com
†   All authors contributed equally to this work.

**Abstract:** This paper demonstrates the ability of a functional food based on probiotics and selenium nanoparticles (SeNPs) to annihilate the toxic effect of cadmium on the kidneys. SeNPs were obtained by eco-friendly method used *Lactobacillus casei*. The morphological features and size of SeNPS were characterized by Atomic Force Microscopy (AFM) and Dynamic Light Scattering (DLS). Two kind of SeNPs were used, purified and Lacto-SeNPs (LSeNPs), administered by gavage at three concentrations (0.1, 0.2, and 0.4 mg/Kg b.w.) for 30 days in a mouse model of cadmium renal toxicity. The blood marker of renal injury (creatinine) significantly decreased in groups where the mice were treated with both form of SeNPs. The antioxidant capacity of plasma was evaluated by Trolox Equivalent Antioxidant Capacity (TEAC) assay and revealed that SeNPs in co-treatment with Cd, promotes maintaining antioxidant activity at the control level. Histopathological analysis of kidneys demonstrated morphological alteration in the group that received only cadmium and restored after administration of SeNPs or LSeNPs. In addition, immunohistochemical analysis revealed anti-apoptotic effects through reduction of pro-apoptotic *bax* and increasing of anti-apoptotic *Bcl-2* protein expressions. Moreover, co-administration of Cd with SeNPs significantly decreased gene expression of kidneys inflammatory markers (TNF-α, IL-6, NF-kB) in a dose dependent manner, with the best results for LSeNPs at highest dose (0.4 mg/kg). Therefore, the *L. casei* strain is a potential SeNPs-enriched probiotic for application as functional food in the future to annihilate cadmium-induced kidneys toxicity.

**Keywords:** cadmium; selenium; *Lactobacillus casei*; kidneys; histology; apoptosis; inflammation; functional food; probiotics

## 1. Introduction

Cadmium (Cd) is a toxic heavy metal and is included by *International Agency for Research on Cancer classification* as carcinogenic to humans (group I) [1]. The sources of Cd contamination are industry, where is used as a corrosive reagents, color pigments,

and especially as batteries, and agriculture where is present as impurity in commercial inorganic fertilizers [1–4].

Cd absorption in the body takes place mainly through respiratory and gastro-intestinal tract (by consumption of contaminated food and water) [5]. In the body, Cd is transported by erythrocytes and albumin in the bloodstream and then accumulated in the kidneys, liver, and gut [6–8]. The critical organ of Cd action is kidneys, where has a lot of cytotoxic and metabolic effects. Cd is accumulated throughout a lifetime, a dangerous feature because it produces renal and hepatic dysfunction, anemia, and cancer in different organs, including the kidneys [9,10].

After exposure to Cd through diet, Cd may be absorbed as complexes with different dietary constituents or with metal-binding protein (metallothionein, MT). MT is cysteine-rich protein, with a low molecular weight and has a protective contribution in Cd-induced nephrotoxicity and hepatotoxicity [9,11–13].

The inorganic or organic chelators of heavy metals (calcium or zinc trisodiumdi-ethylenetriaminepentaacetate; carbodithioates; dimercaprol, etc.) were successfully used in remediation of intoxication with heavy metals [9], but this conventional treatment has several adverse effects such as anemia, mineral deficiency, cardiac arrest, or kidney overload [9]. In the last years, the researchers were focused on probiotics [5,14] and nanoparticles [15] as option for the treatment of heavy metals intoxications. Lactic acid bacteria (LAB) are safe microorganisms (included in GRAS category) and are widely used in food industry to produce especially dairy fermented foods, such as yogurt and cheese, besides fermented vegetables, and meats. In addition, lactic acid bacteria are considered probiotics for human with positive effects on health [5,16]. LAB were studied for their ability in removing heavy metals from environment, mainly from contaminated water [17]. Because LAB have negative surface charge, the cations (Cd, Pb, and As) could be binding at the bacteria surface by ion exchange mechanism [17,18].

*Lactobacillus* species could include selenium (Se) intracellular in the organic form such as selenocysteine [19]. On the other hand, the research studies demonstrated the ability of *Lactobacillus* species to convert inorganic form of selenium in nanoparticles [20–22].

Selenium (Se) is an essential trace element for human [23] and in the same time is a contradictory mineral, because at higher level become toxic for the organism, while its deficiency produce several health problems [24]. The World Health Organization has established for Se a value of 70 μg/day for the maximum daily intake, considering that doses above 400 μg/day may exert toxic actions [25].

The foods, like as meat, cereals, and seafoods are the mainly sources for Se in the human body. The forms of Se include (i) organic such as selenomethionine (in cereals and yeast), selenocysteine (in foods of animal origins); (ii) inorganic, i.e., selenite and selenate (Se in the +4 and +6 oxidation state, respectively), which are present in dietary supplements and water [26,27]; and (iii) elemental SeNPs (in the 0 oxidation state) biosynthesized by different microorganism or plants [20,28].

In this moment, some research study on animals or cell cultures, have investigated and demonstrated that Se supplementation can alleviate the toxic effects generated by Cd [26], but there are few studies on human populations [29]. Chen et al. [29] investigated the renal effects of chronic co-exposure to high levels of Cd and Se of subjects from China's Hubei Province by determining the content of metals from blood, urine, and hair along with urine and blood biomarkers. Their results showed that the populations did not show renal tubular or glomerular injury, which is explained by the role of Se in improving Cd induced nephrotoxicity by activating antioxidant enzymes systems. An effect of selenium (Se) on Cd toxicity was observed in a study of Bangladeshi preschool children, aged 4.4–5.4 years [30]. The measured Cd effects were kidney volume, determined by ultrasonography, and estimated glomerular filtration rate (eGFR) calculated from serum cystatin C levels. Urinary Cd levels were inversely associated with eGFR, especially in girls. A beneficial effect of Se was suggested in a Chinese case-control study that included 240 invasive breast cancer cases and 246 age-matched non-cancer controls [31]. There was a

2.83-fold increase in breast cancer risk in women with urinary Cd in the highest tertile and urinary Se in the lowest tertile [31]. The risk of breast cancer was also reduced in women with urinary Se in the middle tertile [31].

Based on the information mentioned above, the authors focused on finding ecofriendly solutions to alleviate the toxic effects of Cd on kidneys. To date, there are several studies focused on ability of lactic bacteria or different nanoparticles to reduce the toxic effects of heavy metals [14,32,33], whereas only one referred to SeNPs-enriched probiotics in a murine model of Cd-induced renal toxicity [34].

The novelty of our study resulted from the use of endogenous nanoselenium biosynthesizes by lactic acid bacteria (*L. casei*), in order to provide a possible functional food which is able to alleviate the toxic effect of cadmium on kidneys.

In this study, SeNPs biosynthesis was produced by *L. casei* and consequently two products were obtained: purified SeNPs (bacteria was removed by acidic hydrolysis) and Lacto-SeNPs (bacteria enriched in SeNPs). These products where further tested in vivo for their ability to alleviate toxic effects induced by cadmium in kidneys. For this purpose, both SeNPs products were administered orally to mice for 30 days at three different concentrations (0.1, 0.2, and 0.4 mg/kg b.w.) and the creatinine (CREA), total antioxidant plasma (by TEAC assay), along with the histology, immunohistochemistry for mitochondrial apoptosis markers (*bcl-2*, *bax*), and gene expression of hepatic inflammatory markers (NF-kB, TNFα, and IL-6) were analyzed in terms of comparative evaluation of dose-dependent protective activity of SeNPs against cadmium intoxication.

## 2. Materials and Methods

### 2.1. Eco-Friendly Biosynthesis and Characterization of SeNPs

In this work, two nano-selenium forms (purified NanoSelenium called, SeNPs and Lacto-NanoSelenium, called, LSeNPs) were produced according to the patent of Prokisch and Zommara [22].

### 2.1.1. Production of Purified NanoSelenium Particles (SeNPs)

The culture medium was prepared by dissolved 5.5 g of MRS in 100 mL distilled water, then boiling for 30 min at 120 °C. After cooling, in the MRS medium was introduced sodium hydrogen selenite (NaHSeO$_3$) in a final concentration of 200 mg/L. For the inoculation, *Lactobacillus casei* (Lyofast LC4P1, Sacco, Cadorago, Italy) was selected for SeNPs biosynthesis in the anaerobic conditions. The initial pH of medium was around 7 and become around 4 at the end of fermentation process. The reaction was allowed to start in a fermentation bottle into the shaking incubator at 37 °C for 48 h until the specific red color of elemental nano-selenium was achieved (Figure 1a). Then, the medium was centrifugation at 6000 rpm, for 15 min and the pellets were uptake in sterilized distilled water. The mechanism of selenium particles formation is mainly intra-cellular for lactic acid bacteria [20], and to obtain the purified SeNPs, the hydrochloric acid (37%) was used in order to remove the bacteria cell wall. The acidic hydrolysis takes 5 days at room temperature and then, the bacterial cells were removed from the mixture by centrifugation at 6000 rpm, for 15 min, the SeNPs were washed with distillated water followed by repeated centrifugation until its pH becomes 7. The purified SeNPs were vacuum filtering and freeze-drying. The resulted powder was observed under AFM (Agilent 5500 AFM, Agilent Technologies, Santa Clara, CA, USA), using tapping mode with RTESP tip. For DLS assay (ZEN 3690, Malvern Instruments, Malvern, Worcestershire, UK) and zeta potential measurements, the SeNPs powder were re-suspended in distillated water and sonicated during 10 min before each measurement to avoid aggregation.

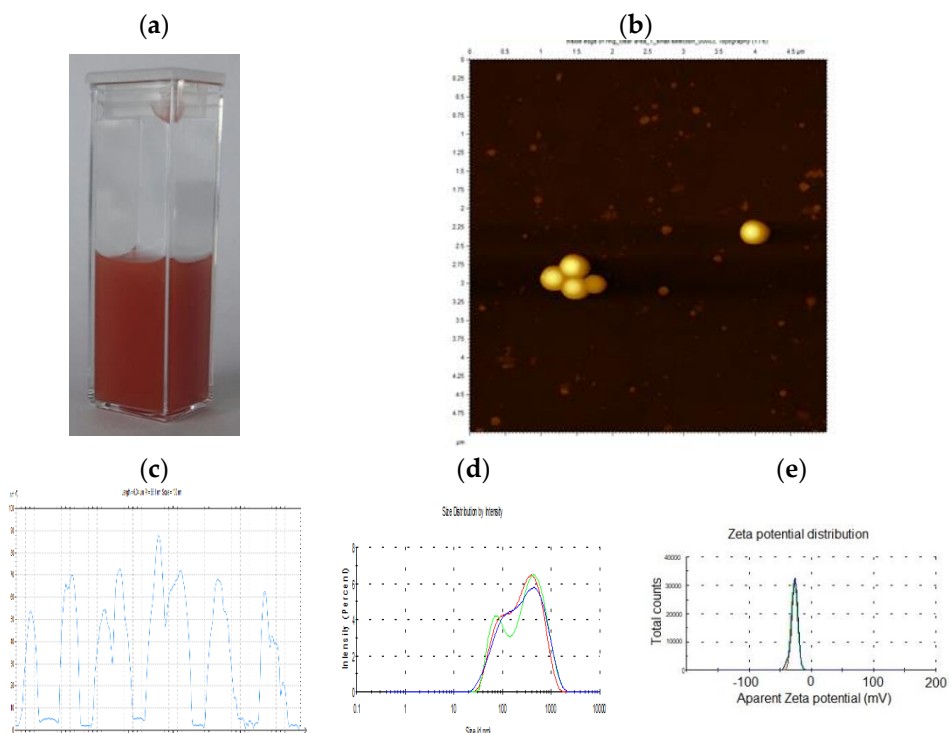

**Figure 1.** (**a**). The characteristic red colour of SeNPs. (**b**). AFM of SeNPs biosynthesized using *L. casei* and NaHSeO₃ as a reducing agent. (**c**). Surface profile of SeNPs recorded from AFM. (**d**). Size distribution measurement of SeNPs. (**e**). Apparent Zeta potential records of SeNPs.

### 2.1.2. Production of Lacto-NanoSelenium Particles (LSeNPs)

The production of LSeNPs was performed as described by Eszenyi et al. [20]. The MRS medium was replaced with skimmed milk and sodium hydrogen selenite (NaHSeO₃) as a reducing agent was used. For the inoculation, *Lactobacillus casei* (Lyofast LC4P1, Sacco) was selected for LSeNPs biosynthesis in the anaerobic condition. Fermentation took place into the shaking incubator at 37 °C for 48 h. At the end of fermentation process, the red/pink yoghurt obtained was centrifuged at 6000 rpm for 15 min and the solid-phase was freeze-drying in order to obtain the powder of LSeNPs.

### *2.2. Animal and Experimental Design*

In the experiment, female CD1 mice (26 ± 3 g) from the University's Animal Facility were used, after prior ethical approval of the working protocol by the Ethical Committee of Vasile Goldis Western University of Arad and certification by the National Sanitary Veterinary and Food Safety Authority of Romania (005/27 February 2017). Mice were fed with an autoclavable standard scientific diet for rodents (Safe D40 diet, SAFE Complete Care Competence, Rosenberg, Germany), which is certified free of toxic substances and balanced regarding the content of amino acids, fatty acids, minerals and vitamins.

Mice were randomly assigned to 8 experimental groups (n = 10):

- Control group—received orally by gavage during the experiment only water;
- Cadmium group (Cd)—received orally 5 mg/kg CdCl₂;
- 0.1 SeNPs + Cd—received orally by gavage 0.1 mg/kg of SeNPs together with 5 mg/kg of CdCl₂
- 0.2 SeNPs + Cd—received orally by gavage 0.2 mg/kg of SeNPs together with 5 mg/kg of CdCl₂
- 0.4 SeNPs + Cd—received 0.4 mg/kg of SeNPs together with 5 mg/kg of CdCl₂
- 0.1 LSeNPs + Cd—received orally by gavage 0.1 mg/kg of LSeNPs together with 5 mg/kg of CdCl₂

- 0.2 LSeNPs + Cd—received orally by gavage 0.2 mg/kg of LSeNPs together with 5 mg/kg of $CdCl_2$
- 0.4 LSeNPs + Cd—received 0.4 mg/kg of LSeNPs together with 5 mg/kg of $CdCl_2$

The three doses of SeNPs (0.1, 0.2, and 0.4 mg/kg b.w.) and route of administration were selected according to the results in which protection was obtained against cadmium, administered to mice at a dose of 5 mg/kg [35].

The administration of NSePs and LNSePs was performed one hour after the administration of Cd. Thirty days after the first oral administration the mice were euthanized under anesthesia with a mixture of ketamine and xylazine. Blood and kidney tissues were collected for further analysis.

### 2.3. Blood Creatinine Level

Venous blood samples were centrifuged at 3500 rpm for 10 min and then analyzed for creatinine (CREA) level (ChemaDiagnostica, Monsano, Italy) with a Mindray BS-120 Chemistry Analyzer (ShenzenMindray Bio-Medical Electronics Co., Ltd., Nanshan, Shenzhen, China).

### 2.4. Antioxidant Capacity of Mice Plasma—TEAC Assay

The Trolox Equivalent Antioxidant Capacity (TEAC) is one of the methods developed to measure the total antioxidant capacity of mice plasma and was performed using modified method of Re et al., 1999 [36]. Shortly, ABTS (2,2′-azinobis (3-ethylbenzothiazoline-6-sulfonic acid) diammonium salt) was dissolved in PBS, pH 7.4 to obtained a 7 mM concentration. The radical cation of $ABTS^{\bullet+}$ solution was produced by reacting with 2.45 mM potassium persulfate, in the dark, at room temperature, overnight (16 h) before use. Before analysis, the radical cation of $ABTS^{\bullet+}$ solution was diluted with PBS, at pH 7.4 in order to obtain an absorbance of $0.70 \pm 0.02$ at 734 nm. After addition of 10 μL of plasma or Trolox standard (final concentration 0–25 μM) to 1 mL of diluted $ABTS^{\bullet+}$, the absorbance was reading exactly 1 min after initial mixing. The results were expressed as percentage inhibition of radical cation of $ABTS^{\bullet+}$.

### 2.5. Histopathology Analysis

Kidney samples were fixed in 4% paraformaldehyde solution in PBS for 24 h, followed by dehydration in a graded series of ethanol, clarified in toluene, following by paraffin embedding. Sections of 5 μm were stained with hematoxylin-eosin (H&E) and examined under an Olympus BX43 light microscope (Tokyo, Japan) and images captured using an XC30 CCD camera (Tokyo, Japan).

### 2.6. Immunohistochemical Analysis

Kidney paraffin embedded sections were deparaffinized in Dewax (Biosystems, Nussloch, Germany) and rehydrated prior to epitope retrieval in Novocastra sol. (Leica Biosystems, Nussloch, Germany). Following neutralization of endogenous peroxidase, sections were incubated overnight at 4 °C with anti *bax* and *bcl-2* antibodies (1:100) (Santa Cruz, Dallas, TX, USA). Detection was then performed using a polymer detection system (cat. no. RE7280 K; Novolink Max Polymer Detection system) and 3,3′diaminobenzidine (DAB) as chromogenic substrate. Nuclei were stained with hematoxylin. Slides were mounted and examined under an Olympus BX43 light microscope.

### 2.7. RT-PCR Analysis

Kidney samples collected on RNAlater (Thermo Fisher Scientific, USA) were stored at −80 °C until processing. The RNA extraction was performed with SV Total RNA Isolation System extraction kit (Promega, Madison, WI, USA), according to the manufacturer's recommendations. The quantitative and qualitative analysis of RNA was assessed by spectrophotometry (NanoDrop 8000, Thermo Fisher Scientific, Waltham, MA, USA). For Real Time PCR, the LuminarisHiGreenqPCT Master Mix kit (Thermo Scientific, Waltham,

MA, USA) and Applied Biosystems 7500 Real Time PCR System (Foster City, CA, USA) were used. Samples were tested in triplicate and the glyceraldehyde 3-phosphate dehydrogenase (GAPDH) gene was used as reference. Primers used were included in Table 1. The results obtained were interpreted using the 2ΔΔCT method of Livak et al., 2001 [37].

**Table 1.** Primer sequences for RT-PCR.

| Target | Sense | Antisense |
|--------|-------|-----------|
| NF-kB 65 | 5′CTTGGCAACAGCACAGACC3′ | 5′GAGAAGTCCATGTCCGCAAT3′ |
| TNF-$\alpha$ | 5′CTGTAGCCCACGTCGTAGC3′ | 5′TTGAGATCCATGCCGTTG3′ |
| IL-6 | 5′AAAGAGTTGTGCAATGGCAATTCT3′ | 5′AAGTGCATCATCGTTGTTCATACA3′ |
| GAPDH | 5′CGACTTCAACAGCAACTCCCACTCTTCC3′ | 5′TGGGTGGTCCAGGGTTTCTTACTCCTT3′ |

*2.8. Statistical Analysis*

All values represent mean $\pm$ standard deviation (SD) for 10 mice in each group, and statistically significant differences (* $p < 0.05$; ** $p < 0.01$; *** $p < 0.001$ and ### $p < 0.001$) were determined compared with control and cadmium group (Cd), respectively. Data were statistically processed using GraphPad Prism 3.03 software (GraphPad Software, Inc., La Jolla, CA, USA), and one-way analysis of variance, followed by a Bonferroni's Multiple Comparison Test.

**3. Results and Discussion**

*3.1. Physico-Chemical Characterization of SeNPs*

The appearance of red colour (Figure 1a) suggest the formation of elemental nano-selenium, which has been confirmed by us through AFM and DLS.

As evidenced by AFM image (Figure 1b), regular, spherical shape, Se nanoparticles were obtained by eco-friendly synthesis. The DLS measurements (Figure 1c) revealed two maximum size distributions: first one with average size of 90 nm and lower concentration, and second one, with average size of about 400 nm and higher concentration. Even SeNPs tends to aggregate into larger size spheres (including at room temperature) due to their high surface to volume ratio, we were able to identify single nanoparticles, well distributed on the mica-support surface, without any aggregation. Moreover, the size of SeNPs obtained by DLS assay was in line with the observed AFM details. The zeta potential measurements (Figure 1d) indicates −26.6 mV, which is considered an indicative for good stability, according to the literature [38]. It is well known that the criteria of stability of NPs are measured when the values of zeta potential ranged from higher than +20 mV to lower than −20 mV [39].

Different microorganisms are able to reduce inorganic selenium into red elemental SeNPs in different size with an unique structured nanospheres with regular and uniform size [20,22,39]. According to previous our work [20] selenium in the LSeNPS is manly (>95%) in a form of nanoselenium and the rest (<5%) is organic selenium.

Xu et al. [40], demonstrated that probiotic *L. casei* 393 is able to transform sodium selenite to SeNPs under anaerobic conditions which is considered to be one of the mechanism of Se detoxification. Three different ways were proposed to explain the bioconversion of Se from +4 to 0 oxidation state [40]: (i) the periplasmic nitrite reductase (nitrite reductase, sulfite reductase, and GSH reductase); (ii) redox precipitation of both elemental sulfur and elemental Se; and (iii) a glutathione reductase catalyzes the reaction of GSH with Se +4 to produce GS–Se–SG, further generate GS–Se.

*3.2. Effect of SeNPs on Blood Creatinine Level*

The results of CREA are shown in Figure 2. The level of CREA is used as an indicator of renal function [41]. The blood CREA level increases after Cd administration by 17%, but the treatment with SeNPs in both forms (purified and LSeNPs) significantly decreased the level of this indicator except for the group were the lowest dose of SeNPs was used

(0.1 mg/kg). The lowest CREA levels was recorded for the 0.2 mg/kg of LSeNPs group, demonstrated the good efficiency of LSeNPs to annihilate the Cd-induced renal toxic effect.

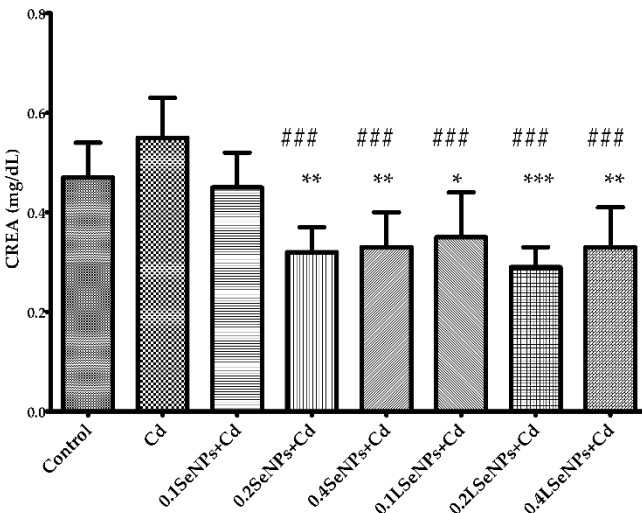

**Figure 2.** Effects of cadmium and both form of selenium (SeNPs, purified and LSeNPs) on CREA level and LDH activity after 30 days treatments. All values were expressed as mean ± SD for 10 mice in each group. Groups: Control-mice received only water; Cadmium group (Cd)-mice received orally 5 mg/kg CdCl$_2$; 0.1 SeNPs + Cd (0.1 mg/kg SeNPs + 5 mg/kg CdCl$_2$); 0.2 SeNPs + Cd (0.2 mg/kg SeNPs + 5 mg/kg CdCl$_2$); 0.4 SeNPs + Cd (0.4 mg/kg SeNPs + 5 mg/kg CdCl$_2$); 0.1 LSeNPs + Cd (0.1 mg/kg LSeNPs + 5 mg/kg CdCl$_2$); 0.2 LSeNPs + Cd (0.2 mg/kg LSeNPs + 5 mg/kg CdCl$_2$); 0.4 LSeNPs + Cd (0.4 mg/kg LSeNPs + 5 mg/kg CdCl$_2$). Groups: SeNPs + Cd and LSeNPs + Cd at different concentration vs. control group: * $p < 0.05$; ** $p < 0.01$; *** $p < 0.001$. Groups: SeNPs + Cd and LSeNPs + Cd at different concentration vs. cadmium group (Cd): ### $p < 0.001$.

No significant differences were observed in CREA level when the male Sprague–Dawley rats were administrated gavage SeNPs at levels between 0.2 and 8.0 mg/kg body weight, demonstrating that SeNPs were not toxic at the concentrations tested [42]. In other study, the CREA levels were higher in male albino rats group that receive Cd compared with the control group, instead not significantly changes were recorded in Se treated group alone or Cd + Se treated group compared with control group [23].

*3.3. Antioxidant Capacity of Mice Plasma—TEAC Assay*

The most commonly methods used for assessing total antioxidant capacity of plasma or serum are FRAP (Ferric Reducing Antioxidant Power) and TEAC (Trolox Equivalent Antioxidant Capacity). The mechanisms of measuring antioxidant capacity are different in these methods. For examples, FRAP assay measures the iron reducing ability of biological sample [43], while TEAC assay is based on inhibition of radical cation ABTS$^{\bullet+}$ (2,2′-azinobis (3-ethylbenzothiazoline 6-sulfonate) inhibition [44].

Antioxidant capacity of mice plasma was evaluated using TEAC assay, where potassium persulfate as radical-initiator of ABTS was used. The concentration–response curve of different Trolox solutions (0.25–2 mM) is shown in Figure 3a. The TEAC values of mice plasma, expressed as % inhibition of ABTS is shown in Figure 3b. Our results revealed significant decrease in the antioxidant capacity of mice plasma for Cd group, while Cd co-treatments with SeNPs did not affect the antioxidant capacity compared to the control group, suggesting selenium protection.

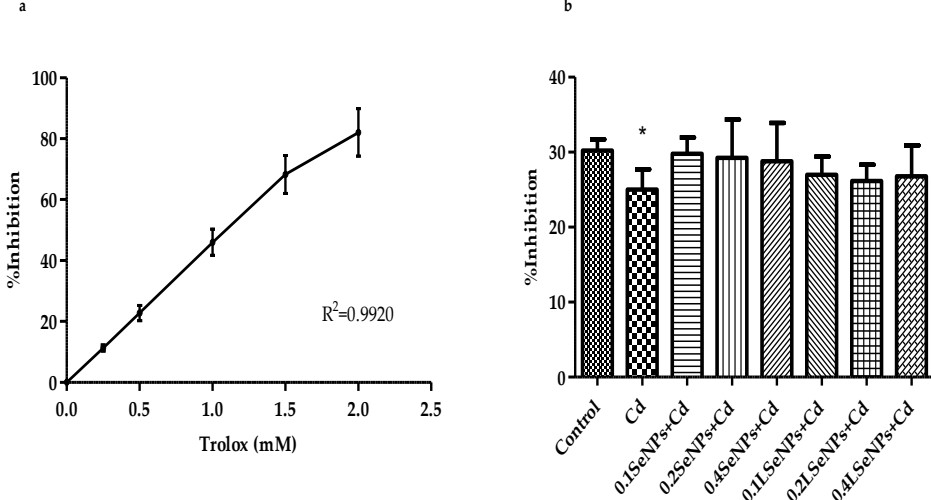

**Figure 3.** (**a**). Trolox concentrations-response curve of % Inhibition of ABTS$^{\bullet+}$, recorded at 734 nm. Five different standard concentrations (between 0.25 and 2 mM Trolox) were used in triplicate. (**b**).The % Inhibition of ABTS$^{\bullet+}$ from blood mice after 30 days treatments. All values were expressed as mean $\pm$ SD for 10 mice in each group. Statistically significant difference at * $p < 0.05$. Groups: Control-mice received only water; Cadmium group (Cd)-mice received orally 5 mg/kg CdCl$_2$; 0.1 SeNPs + Cd (0.1 mg/kg SeNPs + 5 mg/kg CdCl$_2$); 0.2 SeNPs + Cd (0.2 mg/kg SeNPs + 5 mg/kg CdCl$_2$); 0.4 SeNPs + Cd (0.4 mg/kg SeNPs + 5 mg/kg CdCl$_2$); 0.1 LSeNPs + Cd (0.1 mg/kg LSeNPs + 5 mg/kg CdCl$_2$); 0.2 LSeNPs + Cd (0.2 mg/kg LSeNPs + 5 mg/kg CdCl$_2$); 0.4 LSeNPs + Cd (0.4 mg/kg LSeNPs + 5 mg/kg CdCl$_2$).

Total antioxidant capacity depending on the chemical reactions involved can be classified into two classes: hydrogen atom transfer and on single electron transfer methods. The TEAC assay is part of the last method, where the mechanism is based on the ability of a sample to transfer one electron to reduce radicals. The ABTS radical is soluble in both aqueous and organic solvent which allows the simultaneous evaluation of hydrophilic and lipophilic compounds being one of the advantages of TEAC method [43].

Cadmium is unable to generate directly free radicals, but superoxide radical, hydroxyl radical could be generated indirectly [9]. The studies on animals shown the acute intoxication with cadmium increased antioxidant defense enzymes, including catalase, glutathione peroxidase, glutathione-S-transferase and superoxide dismutase [1,9,45]. Our results shows that both form of SeNPs were able to counteract the oxidative stress effects of Cd. Se antagonizes the toxicity of Cd through the action of Se-dependent antioxidant enzymes such as glutathione peroxidase and thioredoxin reductase [26]. Moreover, it is known that selenium pretreatment of rats intoxicated with cadmium led to a significant decrease in MDA concentration, and increased levels of glutathione (GSH) and glutathione peroxidase (GPx) and thioredoxin reductase (TrxR) activities, when compared with those of cadmium-treated group; by this enzymatic mechanism it reduces the renal lipid peroxidation and induced a significant restoring of the antioxidant system affected by cadmium administration [46]. In humans (controlled trial study) selenium might reduce the oxidative stress by increasing total antioxidant capacity (TAC) and glutathione peroxidase (GPX) levels and decreasing serum malonaldehyde (MDA), both of which being crucial factors for reduction of oxidative stress [47].

### 3.4. Histopathology Analysis

Light microscopic examination showed a normal structure of the kidney (Figure 4a) in the controls. Glomerular atrophy, tubular necrosis and widening were evident in the kidneys of all Cd-exposed animals (Figure 4b). Simultaneous administration of SeNPs with Cd, reduced the heavy metal toxic structural changes in the kidney in dose dependent manner. The kidney morphology of the LSeNPs–Cd groups did not differ from that of

the control animals, offering a complete prevention from the Cd-induced changes in renal structure at highest dose (Figure 4h).

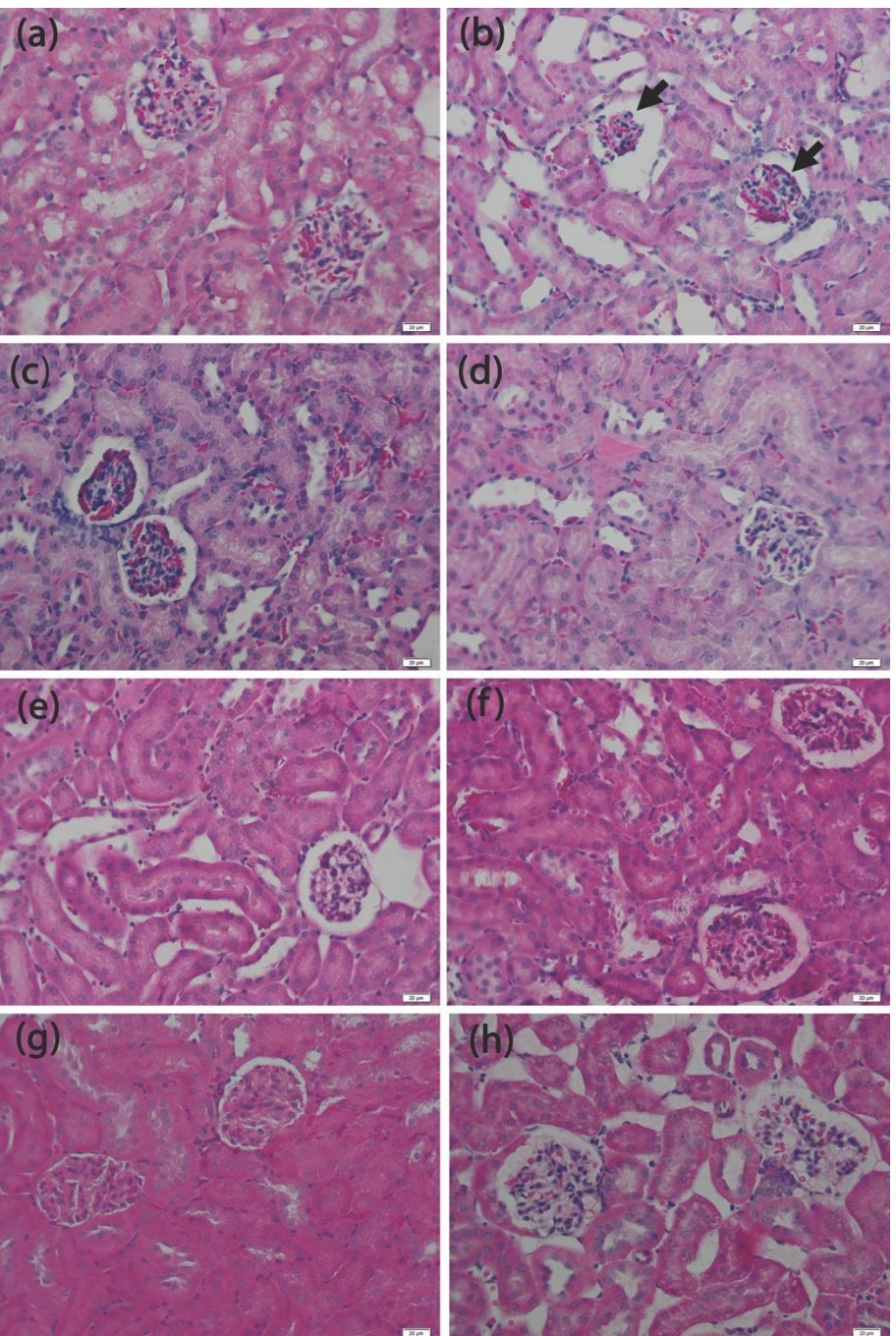

**Figure 4.** Histopathological sections of kidney of the experimental groups, H&E × 200. Various degrees of histopathological changes were observed in Cd exposure groups, mainly including glomerular atrophy (arrow) and tubular damage (**b**). Co-administration of SeNPs with Cd, reduced the metal toxic histological changes in the kidney in dose dependent manner, more obvious for LSeNPs. CV-centrilobular vein. Groups: Control (**a**)-mice received only water; Cadmium group (**b**)-mice received orally 5 mg/kg CdCl$_2$; 0.1 SeNPs + Cd (**c**) (0.1 mg/kg SeNPs + 5 mg/kg CdCl$_2$); 0.2 SeNPs + Cd (**d**) (0.2 mg/kg SeNPs + 5 mg/kg CdCl$_2$); 0.4 SeNPs + Cd (**e**) (0.4 mg/kg SeNPs + 5 mg/kg CdCl$_2$); 0.1 LSeNPs + Cd (**f**) (0.1 mg/kg LSeNPs + 5 mg/kg CdCl$_2$); 0.2 LSeNPs + Cd (**g**) (0.2 mg/kg LSeNPs + 5 mg/kg CdCl$_2$); 0.4 LSeNPs + Cd (**h**) (0.4 mg/kg LSeNPs + 5 mg/kg CdCl$_2$).

Several histopathological studies revealed that the toxic effects of Cd in the kidney are confined with proximal tubular cells, affected by necrosis and apoptosis and tubular degeneration [48–50]. Early tubular disease in Cd nephrotoxicity [51] and tubular proteinuria continued with glomerular damage, leading to albuminuria and a progressive impairment in glomerular filtration, are injuries induced by cadmium, causing end-stage renal failure [52]. In our experiment we noticed several histopathological renal changes, including both of tubular injuries and glomerular atrophy or swelling, which are in agreement with previous toxicological results, suggesting that exposure to Cd 5 mg/kg to mice induced morpho-functional changes and renal dysfunction.

Cd is absorbed from the gastrointestinal tract into the circulation by a transporter, DMT1 (Divalent Metal Ion Transporter 1), in the duodenum [53]. Once it reaches the liver it is rapidly bound to metallothionein (MT), which is then slowly released back into circulation. The Cd-MT complex is filtered at the glomerulus and reabsorbed by the proximal tubule, being a protective response to limit toxicity from free Cd ($Cd^{2+}$). Unfortunately, once the MT-producing capacity of proximal renal tubular cells is exhausted, progressive tubular cell damage occurs and the intracellular levels of $Cd^{2+}$ increase [54]. Cd can also induce autoantibodies to MT, which may be toxic for tubular cells and interfere with Cd detoxification [55], while other autoantibodies target induced glomerular damage [51], and manifested by morphological changes on both tubular and glomerular cells. In our experiments, a dose-dependent increase protection against nephrotoxicity induced by cadmium was obtained with the highest dose of LSeNPs, which might be associated with redox regulatory effect of Se, showed by recovery of glutathione peroxidase and thioredoxin reductase activities, decreasing free radical-mediated lipid peroxidation and glutathione regeneration [56].

### 3.5. The SeNPs Prevent Apoptosis in Renal Parenchyma Induced by Cadmium

The *Bcl-2* family proteins play a crucial role for the mitochondria dependent apoptotic pathway [57]. *Bcl-2* is known as an anti-apoptotic protein which protects the cells from apoptosis, whereas the pro-apoptotic proteins such as *Bax* promote the programmed cell death.

Exposure to Cd causes the activation of apoptosis-related mitochondrial signaling and DNA damage response [58], through unbalancing *Bcl-2/Bax* ratio which determines cell death [59], and considering a valuable tool to assess renal protective effects of SeNPs and LSeNPs against the pro-apoptotic activity of Cd.

As shown in Figure 5, Cd exposure increased immunopositivity of pro-apoptotic *Bax* protein, especially at tubular level and decreased expression of anti-apoptotic *Bcl-2* protein, however, co-treatment with SeNPs and LSeNPs induced a marked reduction in positivity of *Bax* and increased *Bcl-2* protein positivity in a dose-dependent manner.

Mitochondria is the major intracellular target for Cd toxicity, inhibiting directly Na/K pump [60]; accumulation of $Cd^{2+}$ is followed by inhibition of the respiratory chain by acting at the level of complex III [61] and resulting in the generation of reactive oxygen species (ROS) [62], and mitochondrial damage [63] with release of cytochrome c [64], leading to caspase activation, causing cell death by apoptosis and necrosis [65–67]. Moreover, previous studies showed that exposure to Cd down-regulated anti-apoptotic *Bcl-2* protein and up-regulated pro-apoptotic *Bax* protein, suggesting that Cd-induced apoptosis via regulations of *Bcl-2* and *Bax*, while selenium has an ability to inhibit mitochondrial apoptotic pathway in oxidative stress mediated kidney dysfunction caused by cadmium and rebalanced the *Bcl-2/ Bax* ratio [68], as we showed in our experiment.

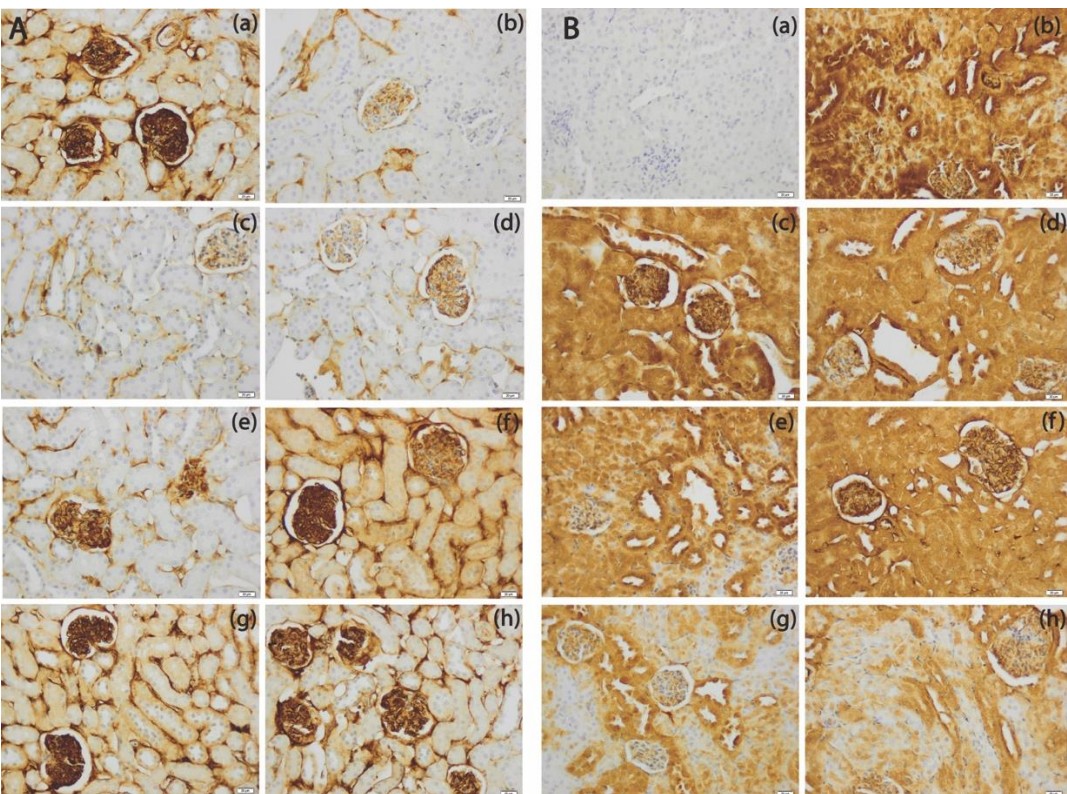

**Figure 5.** Renal immunohistochemical expression of *Bcl-2* (**A**) and *Bax* (**B**) in: Groups: Control (**a**)-mice received only water; Cadmium group (**b**)-mice received orally 5 mg/kg CdCl$_2$; 0.1 SeNPs + Cd (**c**) (0.1 mg/kg SeNPs + 5 mg/kg CdCl$_2$); 0.2 SeNPs + Cd (**d**) (0.2 mg/kg SeNPs + 5 mg/kg CdCl$_2$); 0.4 SeNPs + Cd (**e**) (0.4 mg/kg SeNPs + 5 mg/kg CdCl$_2$); 0.1 LSeNPs + Cd (**f**) (0.1 mg/kg LSeNPs+ 5 mg/kg CdCl$_2$); 0.2 LSeNPs + Cd (**g**) (0.2 mg/kg LSeNPs + 5 mg/kg CdCl$_2$); 0.4 LSeNPs + Cd (**h**) (0.4 mg/kg LSeNPs + 5 mg/kg CdCl$_2$).

### 3.6. The SeNPs Prevent Inflammation in Kidney Parenchyma Induced by Cadmium

Central events in Cd-induced tissue injury include generation of reactive oxygen species, and inflammation with increased production of proinflammatory cytokines leading to renal tissue damage [26,64,69,70]

As shown in Figure 6, Cd exposure up-regulated TNF-α, IL-6, NF-kB, however, co-treatment with SeNPs induced their down-regulation in a dose-dependent manner, highlighted for LSeNPs at highest dose.

In our study we demonstrate that Cd-induced nephrotoxicity is associated with increased renal gene expresion of Nf-kB, TNF- and IL-6, which is consistent with previous reports [71,72]. SeNPs administration significantly and dose-dependently prevented an upregulation in cytokine response in CD-intoxicated mice, while selenite (2 mg kg$^{-1}$ diet to chickens) has the ability to decrease the expression of blood proinflammatory factors (NF-κB and COX-2) that were upregulated following Cd exposure [73].

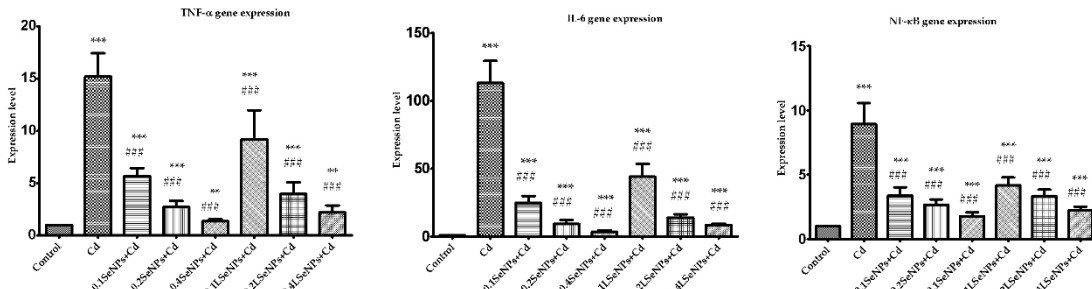

**Figure 6.** Renal TNF-$\alpha$, IL-6, NF-kB gene expressions. Groups: Control-mice received only water; Cadmium group (Cd)-mice received orally 5 mg/kg CdCl$_2$; 0.1 SeNPs + Cd (0.1 mg/kg SeNPs + 5 mg/kg CdCl$_2$); 0.2 SeNPs + Cd (0.2 mg/kg SeNPs + 5 mg/kg CdCl$_2$); 0.4 SeNPs + Cd (0.4 mg/kg SeNPs + 5 mg/kg CdCl$_2$); 0.1 LSeNPs + Cd (0.1 mg/kg LSeNPs + 5 mg/kg CdCl$_2$); 0.2 LSeNPs + Cd (0.2 mg/kg LSeNPs + 5 mg/kg CdCl$_2$); 0.4 LSeNPs + Cd (0.4 mg/kg LSeNPs + 5 mg/kg CdCl$_2$). Groups: SeNPs + Cd and LSeNPs + Cd at different concentration vs. control group: ** $p < 0.01$; *** $p < 0.001$. Groups: SeNPs + Cd and LSeNPs + Cd at different concentration vs. cadmium group (Cd): ### $p < 0.001$.

## 4. Conclusions

Through our study, we validated a new strategy to mitigate the renal toxic effects of cadmium, involving a combination of lactic acid bacteria and SeNPs. To the best of our knowledge, this study is the first attempt to evaluate the protective effects of SeNPs-enriched *L. casei* on in vivo kidneys damage induced by cadmium.

In our study, both selenium products (SeNPs and LSeNPs) have protective effect against Cd induced renal toxicity in dose-dependent manner. Probably, the better effect provided by LSeNPs is due to synergic activity of SeNPs and probiotic bacteria, being known that probiotic bacteria have good ability to bind Cd [18]. In conclusion, Se-enriched *L. casei* could provide a better alternative as a functional food due the double efficacy of Se and probiotics and to be further use to protect against Cd-induced kidneys toxicity.

**Author Contributions:** Conceptualization, A.H., S.I.V. and J.P.; methodology, S.I.V., V.L., A.V.T., C.B., H.H., A.C., S.G., M.R., B.M., L.F., S.C., A.H. and C.C.; formal analysis, S.I.V., V.L., A.V.T., C.B., H.H., A.C., S.G., M.R., B.M., L.F., S.C., C.C., C.P., M.P. and E.M.; resources, S.I.V. and A.H.; writing—review and editing, S.I.V., S.C. and A.H. All authors have read and agreed to the published version of the manuscript.

**Funding:** This research was funded by Romanian Ministry of Research and Innovation, project Number PN-III-P2-2.1-PED-2016-1846.

**Institutional Review Board Statement:** The study was conducted according to the guidelines of the Declaration of Helsinki and approved by the Ethics Committee of Vasile Goldis Western University of Arad (Certificate No. 71/07.06.2016).

**Informed Consent Statement:** Not applicable.

**Data Availability Statement:** The data presented in this study are available on request from the corresponding author. The data are not publicly available due to privacy.

**Acknowledgments:** The support of Traian Octavian Costea from University of Oradea for recording the AFM image is highly acknowledged.

**Conflicts of Interest:** The authors declare no conflict of interest.

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
