# Peer review of "Functional Food Product Based on Nanoselenium-Enriched Lactobacillus casei against Cadmium Kidney Toxicity"

_applsci, doi:10.3390/app11094220_

Round 1

Reviewer 1 Report

Overall comments

In a paper entitled, “Functional food product based on nanoselenium-enriched Lactobacillus casei against cadmium kidney toxicity”, the authors attempted to mitigate nephrotoxicity of cadmium using two forms of selenium as purified nano-selenium (SeNPs) and lyophilized lactobacillus nano-selenium (LSeNPs).  Each form of selenium was administered in three dose levels (0.1, 0.2 and 0.4 mg/kg) to mice via gavage route together with Cd as CdCl2 at a dose of 5 mg/kg body weight for 30 days. 

This reviewer finds many shortcomings in this paper that include factual errors, a lack of clear hypothesis and relevant literature especially in humans.  Most references are apparently irrelevant.  The most concerning issues are a study design and fundamental knowledge on biochemistry/nutrition/toxicity of selenium as detailed below.

Specific comments

  1. Line 48, “Cd was used in agriculture as chemically fertilizers”. This is incorrect. Cd is a contaminant or impurity in phosphate fertilizers used in agriculture.
  2. Lines 49, 50. “Cd absorption in the body takes place through respiratory, gastro-intestinal tract and by skin”. Skin is NOT a route of Cd entry to the body (see Toxicological Profile for Cadmium, Agency for Toxic Substances and Disease Registry).
  3. A hypothesis should be explicitly stated. The authors seemed to suggest that nanoselenium may have chelating propensities, but data on Cd accumulation in kidneys/livers were provided to support or refuse this hypothesis. Instead, the authors provided data on histopathology of kidneys and an overall kidney function measure as plasma creatinine concentration.
  4. Literature reports of protective effects of selenium against Cd toxicity in kidneys of humans should be provided.
  5. Experiments should include also selenium alone (inorganic or any organic selenium compounds such as selenoneine with known biological effects). The authors referred to reference #23 to explain their choices of selenium/cadmium dose levels.  However, it is noteworthy the reference cited reported effect of selenium on a different target (spermatogenesis in testes), not kidneys.

Ref # 23: Ren XM, Wang GG, Xu DQ, Luo K, Liu YX, Zhong YH, Cai YQ. The Protection of Selenium on Cadmium-Induced Inhibition of Spermatogenesis via Activating Testosterone Synthesis in Mice. Food Chem Toxicol 2012, 50, 3521–3529.

  1. The author should provide information concerning recommended dietary allowance for selenium and its toxic levels.
  2. The authors offered no explanations as to why only LSeNPs at highest dose produced a better effect than SeNPs?
  3. There is no information regarding what forms of selenium present in the LSeNPs. This is important in selenium research. For example, selenoneine that is present in bluefin tuna has been found to be a potent antioxidant as reported in two papers below.

Yamashita Y, Yabu T, Yamashita M. Discovery of the strong antioxidant selenoneine in tuna and selenium redox metabolism. World J Biol Chem. 2010 May 26;1(5):144-50.

Yamashita Y, Yamashita M.  Identification of a novel selenium-containing compound, selenoneine, as the predominant chemical form of organic selenium in the blood of bluefin tuna.J Biol Chem. 2010 Jun 11;285(24):18134-8.

Author Response

Response to Reviewer 1

The authors would like to thanks for Reviewer 1 comments that improve our manuscript quality. 

The manuscript was checked for type-errors and English reviewed.

Overall comments

In a paper entitled, “Functional food product based on nanoselenium-enriched Lactobacillus casei against cadmium kidney toxicity”, the authors attempted to mitigate nephrotoxicity of cadmium using two forms of selenium as purified nano-selenium (SeNPs) and lyophilized lactobacillus nano-selenium (LSeNPs).  Each form of selenium was administered in three dose levels (0.1, 0.2 and 0.4 mg/kg) to mice via gavage route together with Cd as CdCl2 at a dose of 5 mg/kg body weight for 30 days. 

This reviewer finds many shortcomings in this paper that include factual errors, a lack of clear hypothesis and relevant literature especially in humans.  Most references are apparently irrelevant.  The most concerning issues are a study design and fundamental knowledge on biochemistry/nutrition/toxicity of selenium as detailed below.

Specific comments

  1. Line 48, “Cd was used in agriculture as chemically fertilizers”. This is incorrect. Cd is a contaminant or impurity in phosphate fertilizers used in agriculture.

Thank you very much for your comment. We have expressed wrong and we rephrased as:

The sources of Cd contamination are industry, where is used as a corrosive reagents, color pigments and especially as batteries, and agriculture where is present as impurity in commercial inorganic fertilizers.

  1. Lines 49, 50. “Cd absorption in the body takes place through respiratory, gastro-intestinal tract and by skin”. Skin is NOT a route of Cd entry to the body (see Toxicological Profile for Cadmium, Agency for Toxic Substances and Disease Registry).

We removed the words regarding to the absorption of Cd by skin.

  1. A hypothesis should be explicitly stated. The authors seemed to suggest that nanoselenium may have chelating propensities, but data on Cd accumulation in kidneys/livers were provided to support or refuse this hypothesis. Instead, the authors provided data on histopathology of kidneys and an overall kidney function measure as plasma creatinine concentration.

We changed the hypothesis in Introduction Chapter.

  1. Literature reports of protective effects of selenium against Cd toxicity in kidneys of humans should be provided.

There are some research studies on animals or cell cultures (18 experimental reports, Zwolak, 2020)   that investigated and demonstrated that Se supplementation in the form of Na2SeO3 and Na2SeO4 can alleviate the toxic effects generated by Cd, but there are few studies on human populations. However in Introduction chapter, we mention the recent results of Chen et al. 2020.

Zwolak I. The Role of Selenium in Arsenic and Cadmium Toxicity: An Updated Review of Scientific Literature. Biol Trace Elem Res. 2020, 193, 44–63, doi:10.1007/s12011-019-01691-w

Chen J., He W., Zhu X., Yang S., Yu T., Ma W. Epidemiological study of kidney health in an area with high levels of soil cadmium and selenium: Does selenium protect against cadmium-induced kidney injury? Science of the Total Environment 698 (2020) 134106. https://doi.org/10.1016/j.scitotenv.2019.134106

  1. Experiments should include also selenium alone (inorganic or any organic selenium compounds such as selenoneine with known biological effects). The authors referred to reference #23 to explain their choices of selenium/cadmium dose levels.  However, it is noteworthy the reference cited reported effect of selenium on a different target (spermatogenesis in testes), not kidneys.

Ref # 23: Ren XM, Wang GG, Xu DQ, Luo K, Liu YX, Zhong YH, Cai YQ. The Protection of Selenium on Cadmium-Induced Inhibition of Spermatogenesis via Activating Testosterone Synthesis in Mice. Food Chem Toxicol 2012, 50, 3521–3529.

The UVVG Ethics Commission recommended to reduce the number of animals in order to comply with the 3R rule in animal research. Therefore, we chose not to introduce the selenium alone group, having enough data from the literature (Zhang et al., 2008; Yazdi et al.,2012; Hassan &Webster, 2016). This study compared the SeNPs produced by lactic bacteria (purified and in mixture with bacteria). We used the article on spermatogenesis as references (no.23), because it uses the oral route of administration for Se in 3 doses. In the other articles, Se was introduced in the diet and in this way we did not have a control of the administered dose (dose-depending experiment).

Zhang, J.; Wang, X.; Xu, T. Elemental Selenium at Nano Size (Nano-Se) as a Potential Chemopreventive Agent with Reduced Risk of Selenium Toxicity: Comparison with Se-Methylselenocysteine in Mice. Toxicological sciences 2008, 101, 22–31, doi:10.1093/toxsci/kfm221.

Yazdi, M.H.; Mahdavi, M.; Varastehmoradi, B.; Faramarzi M.A.; Shahverdi, A.R. The Immunostimulatory Effect of Biogenic Selenium Nanoparticles on the 4T1 Breast Cancer Model: An in Vivo Study. Biol. Trace Elem. Res. 2012, 149, 22–28, doi:10.1007/s12011-012-9402-0.

Hassan, C.E.; Webster, T.J. The Effect of Red-Allotrope Selenium Nanoparticles on Head and Neck Squamous Cell Viability and Growth. Int. J. Nanomedicine 2016, 11, 3641–3654, doi:10.2147/IJN.S105173.

  1. The author should provide information concerning recommended dietary allowance for selenium and its toxic levels.

We introduced in the Introduction Chapter the following: Selenium (Se) is an essential trace element for human [23] and in the same time is a contradictory mineral, because at higher level become toxic for the organism, while its deficiency produce several health problems [24]. The World Health Organization has established for Se a value of 70 μg/day for the maximum daily intake, considering that doses above 400 μg/day may exert toxic actions [25].

  1. The authors offered no explanations as to why only LSeNPs at highest dose produced a better effect than SeNPs?

In our study, both selenium products (SeNPs and LSeNPs) has protective effect against Cd induced renal toxicity in dose-dependent manner. Probably, the better effect provided by LSeNPs is due to synergic activity of SeNPs and probiotic bacteria, being known that probiotic bacteria have good ability to bind Cd (Ibrahim et al., 2016). In conclusion, Se-enriched L. casei could provide a better alternative as a functional food due the double efficacy of Se and probiotics. The dose depend efficiency was obtain for both SeNPs and LSeNPs.

Se-enriched probiotic bacteria could provide a better alternative as a dietary supplement due to

the double efficacy of Se and probiotics (Xu et al., 2018).

Ibrahim, F., Halttunen, T., Tahvonen, R. & Salminen, S. Probiotic Bacteria as Potential Detoxification Tools: Assessing Their Heavy Metal Binding Isotherms. Can. J. Microbiol. 2016, 52, 877–885, doi:10.1139/w06-043.

Xu C, Guo Y, Qiao L, Ma L, Cheng Y, Roman A. Biogenic Synthesis of Novel Functionalized Selenium Nanoparticles by Lactobacillus casei ATCC 393 and Its Protective Effects on Intestinal Barrier Dysfunction Caused by Enterotoxigenic Escherichia coli K88. Front. Microbiol. 2018. 9:1129. doi: 10.3389/fmicb.2018.01129

  1. There is no information regarding what forms of selenium present in the LSeNPs. This is important in selenium research. For example, selenoneine that is present in bluefin tuna has been found to be a potent antioxidant as reported in two papers below.

The appearance of red colour (Fig. 1a) suggest the formation of elemental nanoselenium, which has been confirmed by us through AFM and DLS.

Moreover, according to our previous work (Eszenyi et al., 2011) selenium in the LSeNPS is mainly (>95%) in a form of nanoselenium and the rest (<5%) is organic selenium.

The papers suggested are very interesting, but the topic are related to the organic form of selenium, Selenoneine where selenium atom is included in the imidazole ring of trimethyl-L-histidine. Our study is related to the red elemental selenium in the form of nanoparticles with average size of 90 nm (at lower concentration), and 400 nm (at higher concentration).

Wang, H.; Zhang, J.; Yu, H. Elemental Selenium at Nano Size Possesses Lower Toxicity without Compromising the Fundamental Effect on Selenoenzymes Comparison with Selenomethionine in Mice. Toxicol Sci 2007, 101, 22–31, doi:10.1016/j.freeradbiomed.2007.02.013

Eszenyi P.,  Sztrik A., Babka B.,  Prokisch J. Elemental, Nano-Sized (100-500 Nm) Selenium Production by Probiotic Lactic Acid Bacteria. International Journal of Bioscience, Biochemistry and Bioinformatics 2011, 1, 148–152.

Yamashita Y, Yabu T, Yamashita M. Discovery of the strong antioxidant selenoneine in tuna and selenium redox metabolism. World J Biol Chem. 2010 May 26;1(5):144-50.

Yamashita Y, Yamashita M.  Identification of a novel selenium-containing compound, selenoneine, as the predominant chemical form of organic selenium in the blood of bluefin tuna.J Biol Chem. 2010 Jun 11;285(24):18134-8.

Submission Date

Reviewer 2 Report

The Authors presented very interesting and reliable results. However some points are necessary to improve:

  1. Editing - some problems with spaces (no spaces before the parentheses in many lines), also please remove years, the parentheses should not be bolded.
  2. Line 99 - shloud be replaced
  3. Section 2.1 is unclear and should be rephrased. Please specify fermentation condidtions (aerobically/anaerobically; static/shaking?). Bacteria were removed by centrifugation - how was SeNPs pellet obtained?  Why were LSePs not purified of bacteria when produced? 
  4. What were the mice fed during the 30 day period? 
  5. The quality of Figures 1c,d,e is to low.
  6. Line 214 - Unclear sentence - L. casei suggests that bacteria annihilated Cd toxic effect - please rephrase.
  7. Was cadmium bound in the kidneys and not toxic by the use of SeNPs or was it not bound and excreted in the urine? Please comment. 
  8. Please use italics in latin names (section References).
  9. Reference 19 - please add patent number.
  10. References should be unified - commas / semicolons / dots. 

Author Response

Response to Reviewer 2

The authors would like to thanks for Reviewer 1 comments that improve our manuscript quality. 

Comments and Suggestions for Authors

The Authors presented very interesting and reliable results. However some points are necessary to improve:

  1. Editing - some problems with spaces (no spaces before the parentheses in many lines), also please remove years, the parentheses should not be bolded.

The manuscript was checked for type-errors and the problems were solved.

  1. Line 99 - shloud be replaced

The section 2.1. was modified.

  1. Section 2.1 is unclear and should be rephrased. Please specify fermentation condidtions (aerobically/anaerobically; static/shaking?). Bacteria were removed by centrifugation - how was SeNPs pellet obtained?  Why were LSePs not purified of bacteria when produced? 

The section 2.1. was modified according with your suggestion.

  1. What were the mice fed during the 30 day period? 

Mice were fed, during the 30 day of experiment  with an autoclavable standard scientific diet for rodents (Safe D40 diet, SAFE Complete Care Competence, Germany), which is certified free of toxic substances and balanced regarding the content of amino acids, fatty acids, minerals and vitamins.

  1. The quality of Figures 1c,d,e is to low.

The figures were replaced and also submitted separately, with better resolution.

  1. Line 214 - Unclear sentence - L. casei suggests that bacteria annihilated Cd toxic effect - please rephrase.

Thank you for your suggestion. We reformulated the sentence: The lowest CREA levels was recorded for the 0.2 mg/kg of LSeNPs group, demon-strated the good efficiency of LSeNPs to annihilate the Cd-induced renal toxic effect.

  1. Was cadmium bound in the kidneys and not toxic by the use of SeNPs or was it not bound and excreted in the urine? Please comment. 

We considered that SeNPs are responsible for the alleviation of Cd  toxicity into kidneys, and not due to urinary excretion. In defense of this statement is the renal toxicity marker (creatinine) which significantly higher for the CD group, suggesting the presence of the toxicant at the tissue level.

  1. Please use italics in latin names (section References).

We modified in the references.

  1. Reference 19 - please add patent number.

The patent number was added to the reference.

  1. References should be unified - commas / semicolons / dots. 

We used the Zotero for the writing the references and the problems were resolved.

Round 2

Reviewer 1 Report

The Authors have addressed issues raised in responses to comments from the Reviewer and in the text of a revised version of a manuscript.  However, with respect to studies in humans suggesting protective effects of selenium, the authors should include below two references in a final version of their paper.  I thank the authors for their hard work.

Skröder, H., Hawkesworth, S., Kippler, M., El Arifeen, S., Wagatsuma, Y., Moore, S.E., et al. 2015. Kidney function and blood pressure in preschool-aged children exposed to cadmium and arsenic--potential alleviation by selenium. Environ. Res. 140:205–213.

An effect of selenium (Se) on Cd toxicity was observed in a study of Bangladeshi preschool children, aged 4.4–5.4 years (Skröder et al., 2015). The measured Cd effects were kidney volume, determined by ultrasonography, and eGFR calculated from serum cystatin C levels. Urinary Cd levels were inversely associated with eGFR, especially in girls.

Wei, X.L., He, J.R., Cen, Y.L., Su, Y., Chen, L.J., Lin, Y., et al. 2015. Modified effect of urinary cadmium on breast cancer risk by selenium. Clin. Chim. Acta 438:80–85.

A beneficial effect of Se was suggested in a Chinese case-control study that included 240 invasive breast cancer cases and 246 age-matched non-cancer controls (Wei et al., 2015). There was a 2.83-fold increase in breast cancer risk in women with urinary Cd in the highest tertile and urinary Se in the lowest tertile (Wei et al., 2015). The risk of breast cancer was also reduced in women with urinary Se in the middle tertile (Wei et al., 2015).

Author Response

Reviewer 1

Thank you very much for your work that improved our manuscript.

Open Review

Comments and Suggestions for Authors

The Authors have addressed issues raised in responses to comments from the Reviewer and in the text of a revised version of a manuscript.  However, with respect to studies in humans suggesting protective effects of selenium, the authors should include below two references in a final version of their paper.  I thank the authors for their hard work.

Skröder, H., Hawkesworth, S., Kippler, M., El Arifeen, S., Wagatsuma, Y., Moore, S.E., et al. 2015. Kidney function and blood pressure in preschool-aged children exposed to cadmium and arsenic--potential alleviation by selenium. Environ. Res. 140:205–213.

An effect of selenium (Se) on Cd toxicity was observed in a study of Bangladeshi preschool children, aged 4.4–5.4 years (Skröder et al., 2015). The measured Cd effects were kidney volume, determined by ultrasonography, and eGFR calculated from serum cystatin C levels. Urinary Cd levels were inversely associated with eGFR, especially in girls.

Wei, X.L., He, J.R., Cen, Y.L., Su, Y., Chen, L.J., Lin, Y., et al. 2015. Modified effect of urinary cadmium on breast cancer risk by selenium. Clin. Chim. Acta 438:80–85.

A beneficial effect of Se was suggested in a Chinese case-control study that included 240 invasive breast cancer cases and 246 age-matched non-cancer controls (Wei et al., 2015). There was a 2.83-fold increase in breast cancer risk in women with urinary Cd in the highest tertile and urinary Se in the lowest tertile (Wei et al., 2015). The risk of breast cancer was also reduced in women with urinary Se in the middle tertile (Wei et al., 2015).

Thank you very much for the references regarding to protective effects of selenium in humans. These references were included in the manuscript.

Thank you for appreciation our work!

Reviewer 2 Report

I reanalyzed the document and the answers given by the authors, and consider that the document was clearly improved. Just 2 very minor corrections

  1. Line 132 - please use italics for lactobacillus casei
  2. Lines 250 and 306 - * instead of #

Author Response

Reviewer 2

Thank you very much for your work that improved our manuscript.

Open Review

Comments and Suggestions for Authors

I reanalyzed the document and the answers given by the authors, and consider that the document was clearly improved. Just 2 very minor corrections

  1. Line 132 - please use italics for lactobacillus casei

We used italic for Lactobacillus casei.

  1. Lines 250 and 306 - * instead of #

We used * (p<0.05 ) when the mice groups were compared with control group, and ### when  (p <0.001) mice groups were compared with cadmium group (Cd).

Submission Date

02 April 2021

Date of this review

28 Apr 2021 10:15:12
